# Embelin and Its Derivatives: Design, Synthesis, and Potential Delivery Systems for Cancer Therapy

**DOI:** 10.3390/ph15091131

**Published:** 2022-09-09

**Authors:** Michael Danquah

**Affiliations:** Department of Pharmaceutical Sciences, Chicago State University, 9501 South King Drive, Chicago, IL 60628, USA; mdanquah@csu.edu

**Keywords:** embelin, embelin derivatives, XIAP inhibitor, XIAP, drug delivery, cancer therapy, anticancer

## Abstract

Embelin is a naturally occurring benzoquinone that inhibits the growth of cancer cells, making it a potent anticancer drug. However, the low water solubility of embelin restricts its clinical applicability. This review provides a concise summary and in-depth analysis of the published literature on the design and synthesis of embelin derivatives possessing increased aqueous solubility and superior therapeutic efficacy. In addition, the potential of drug delivery systems to improve the anticancer capabilities of embelin and its derivatives is discussed.

## 1. Introduction

Cancer continues to be one of the most debilitating diseases in the world. Approximately 1.9 million new cases and more than 600,000 cancer-related mortality are anticipated in the United States alone in 2022 [1]. This is despite the tremendous advances in the current treatment modalities such as surgery, radiation, and chemotherapy. Among treatment approaches, chemotherapy is the most widely used. Hence, much effort has been devoted to developing highly potent chemotherapeutic drugs. However, despite these advances, a significant drawback with current anticancer drugs is that they kill both cancerous and healthy cells and ultimately succumb to multidrug resistance (MDR) [2,3]. Therefore, the need for new potent anticancer drugs that are nontoxic to patients and do not succumb to MDR is urgent. Consequently, there has been a growing interest in plant extracts with anticancer properties, as these are likely to have fewer adverse effects [4].

For centuries, the fruit of the *Embelia ribes* Burm. f. plant (Myrsinaceae) (known as false black pepper in English and a few Indian languages) has been employed for treating a variety of infirmities in Ayurveda. These include fever, inflammatory diseases, gastrointestinal disorders, heart and urinary ailments, and central nervous system disorders [5]. The pharmacological properties of the *Embelia ribes* fruit have been attributed to embelin (2,5-dihydroxy-3-undecyl-1,4-benzoquinone) [Figure 1 (**1**)], the active ingredient that was first isolated more than forty years ago and subsequently chemically synthesized [6,7]. Embelin has a broad spectrum of pharmacological properties such as antioxidant, anti-inflammatory, anticonvulsant, antifertility, anti-implantation, hepatoprotective, analgesic, wound-healing, and antibacterial activities [8,9,10,11,12,13,14,15]. However, the anticancer activity of embelin has recently attracted great interest, as evidenced by the several in vitro and in vivo studies on the anticancer properties of embelin reported in the literature [10,16,17,18,19,20,21,22,23,24,25,26,27]. Nonetheless, most of the reported studies focus on the biological effects of embelin instead of developing ways of improving its aqueous solubility and synthesizing more potent analogs.

This review summarizes and analyzes the available literature on the synthesis and derivatization of embelin and highlights the potential of drug delivery platforms to enhance the anticancer properties of embelin. 

## 2. Synthesis of Embelin Derivatives

Several derivatives of embelin have been synthesized based on different biological activities to obtain increased aqueous solubility and superior therapeutic efficacy [28,29,30,31,32,33,34,35,36,37,38,39,40,41,42]. In addition, various synthetic strategies have been proposed and investigated [28,29,37,38,39,41,42]. These approaches chiefly rely on modification of its hydrophobic tail and are briefly discussed below.

Using embelin as a lead compound, Chen and coworkers developed a new class of anticancer drugs to target the X-linked inhibitor of apoptosis protein (XIAP) [42]. Chen et al. reported a synthetic method shown in Figure 1. A 1:1 equivalent of n-butyllithium is used to treat triphenylphosphonium bromide and subsequently reacted with the aldehyde **8** and hydrogenated to give **9**. Oxidation of **9** with ceric ammonium nitrate results in **10**, which is treated with perchloric acid and hydrochloric acid to yield **11**.

In their study, modifications were made only to the hydrophobic tail of embelin. Their strategy was grounded in the theory that the hydrophilic dihydroxyquinone core formed hydrogen bonds with XIAP while the hydrophobic tail interacted with the hydrophobic pocket where the isoleucine residues in the AVPI Smac peptide bind. Modifications ranged from a substituted hydrogen molecule to the more aromatic phenylethyl group. In addition, Chen et al. strategically explored the importance of the C_11_H_23_ long hydrophobic tail and the effect of interaction between a terminal phenyl ring and XIAP. Their findings revealed that the long alkyl hydrophobic tail and terminal phenyl ring significantly affected the binding affinity of XIAP BIR3 and, consequently, the potency in inhibiting cancer cell proliferation. Furthermore, they demonstrated that the length of the hydrophobic chain played a crucial role in embelin’s interaction with XIAP. It is noteworthy that one of the seven derivative molecules (2,5-dihydroxy-3-{2-[4-(2-*m*-tolylethyl)phenyl]ethyl}-1,4-benzoquinone) [Figure 1 (**2**)] reported by Chen and coworkers had a binding affinity of 0.18 µM and was twice as potent as embelin. Compound **2** was also potent in inhibiting the proliferation of MDA-MB-231 and PC-3 cell lines with IC_50_ values of 5 µM and 5.5 µM, respectively.

Until the work of Lamblin et al., the effect of altering the polarity of the linear chain of embelin on its aqueous solubility, biodisponibility, and anticancer activity was unknown. To address this, the authors synthesized a series of hydrophilic derivatives by introducing an amine function on a short carbon chain for subsequent ligation with an amino acid group [43]. However, their synthetic approach maintained the hydrophobic side chain necessary for embelin’s proapoptotic activity.

Lamblin et al. reported a synthetic approach shown in Figure 2. Briefly, oxidation of the phenols **12** was achieved using diacetoxyiodo benzene. Hydrolysis of the methoxy of the resulting benzoquinones moieties **13** and Boc deprotection of the amine were accomplished using hydrochloric acid to obtain **14**. 

Lamblin and coworkers evaluated the embelin derivatives, for example, [Figure 1 (**3**)], and found them to be more hydrophilic than embelin. However, none of the derivatives displayed cytotoxic activity in human epithelial carcinoma KB cancer cells. On the contrary, embelin exhibited an IC_50_ of 5.58 µM and was potent in inhibiting KB cell proliferation. Furthermore, the increased side chain polarity of the embelin derivatives results in poor cellular uptake. The resultant structure–activity relationship insights from this study could lead to the preparation of amphiphilic analogs that are both hydrophilic and potent anticancer agents.

Viault et al. synthesized embelin derivatives by directly linking aromatic groups to the benzoquinone core [39]. The authors reported a synthetic method shown in Figure 3. Phenol **15** was reacted with benzoyl chloride and triethylamine in dichloromethane at 0 °C to yield compound **4**.

Like Chen and coworkers, Viault et al. kept the dihydroxybenzoquinone core unchanged. They modified the nature of the hydrophobic chain by incorporating aromatic groups in the vicinal position of the benzoquinone core [39,42]. Unlike Chen and coworkers, the synthetic method of Viault et al. involved synthesizing two generations of derivative molecules. They used bromobenzoquinone as a pivotal intermediate that was subsequently subjected to Suzuki–Miyaura coupling reactions with various functionalized aromatic boronic acids to produce the first generation of derivative molecules. This approach allowed considerable flexibility in the nature and length of the substituent chain and, consequently, the molecular diversity of the embelin derivatives synthesized. For example, Viault et al. introduced a chain with an ester function in one of their embelin derivatives using 3-(3-hydroxyphenyl)-2,5-dimethoxy-1,4-benzoquinone as the precursor. The resulting 3-(2,5-dimethoxy-3,6-dioxocyclohexa-1,4-dien-1-yl)phenyl benzoate derivative [Figure 1 (**4**)] had a 71% yield. Although Viault et al. synthesized several derivatives, they did not report data from biological tests. Therefore, it is not clear how these modifications affected the therapeutic efficacy.

Martin-Acosta and coworkers synthesized novel dihydro-1*H*-pyrazolo [1,3-*b*]pyridine and pyrazolo[1,3-*b*]pyridine embelin analogs with the primary goal of developing embelin derivatives with superior anticancer activity [28]. The rationale for their synthetic strategy, derived from the reported literature suggesting moieties obtained by fusing pyrazoles to other heterocycles such as dihydro-1*H*-pyrazolo[1,3-*b*]pyridines and pyrazolo[3,4-*b*]pyridines, displayed biological activities. Embelin derivatives were synthesized using a condensation reaction involving embelin, 4-nitrobenzaldehyde, and 3-phenyl-5-aminopyrazole. Martin-Acosta et al. reported a synthetic method shown in Figure 4. Compound **16**, 4-(trifluoromethyl)-benzaldehyde **17**, and 3-amino-5-phenylpyrazole **18** were dissolved in 2 mL dichloroethane and treated with 10 mol% EDDA in a MW tube. The MW tube was sealed, and the reaction mixture irradiated at 150 °C for 10 min and purified by filtration to furnish compound **19**.

As part of their study, Martin-Acosta et al. examined the effect of using different aromatic, heteroaromatic, and aliphatic aldehydes on yield and anticancer effect in eight cancer cell lines. Their data revealed that the embelin derivative with 4-NO_2_ substitution showed the best cytotoxic activity (0.70 ± 0.14 µM) in human promyelocytic leukemia HL-60 cancer cells, while embelin derivatives with the 4-Cl, 4-Br, 4-F, and 4-CF_3_ substitutions exhibited good cytotoxic activity in HEL, K-562, and HL-60 leukemia cells. However, the embelin derivative with the 4-CF_3_ substitution {6-hydroxy-3-phenyl-4-(4-(trifluoromethyl)phenyl)-7-undecyl-1*H*-pyrazolo[3,4-*b*]quinoline-5,8(4*H*,9*H*)-dione} [Figure 1 (**5**)] had the best IC_50_ value (1.00 ± 0.42 µM) in acute erythroid leukemia (HEL) cells. Therefore, it was selected as the candidate molecule for further modification.

Martin-Acosta and coworkers also studied the effect of altering the side chain length of {6-hydroxy-3-phenyl-4-(4-(trifluoromethyl)phenyl)-7-undecyl-1*H*-pyrazolo[3,4-*b*]quinoline-5,8(4*H*,9*H*)-dione} on cytotoxicity activity. Derivatives with the following alkyl side chains were synthesized and studied: (CH_2_)_7_CH_3_, (CH_2_)_5_CH_3_, (CH_2_)_3_CH_3_, and CH_2_CH_3_. Shortening the side chain length resulted in a loss of cytotoxic activity. This observation is consistent with the literature, where embelin derivatives with longer alkyl side chains displayed better cytotoxic activity [42].

Using the crystal structure of PAI-1 in complex with embelin, Chen et al. designed and synthesized a library of embelin analogs to investigate the structure–activity relationships [38]. The authors reported a synthetic approach shown in Figure 5. Compound **20** was treated with *n*-butyllithium and the respective bromide to yield compound **21** which was subsequently treated with 1,4-dioxane and hydrochloric to furnish compound **22**

In their study, modifications were made to the benzoquinone core’s C2, C3, C5, and C6 positions. Following the introduction of methyl groups to the positions C2 and C5, they found that the hydroxyl groups present at the locations C2 and C5 are crucial to maintaining the inhibitory potency of the analogs. They theorize this is the case since both the C2 and C5 positions participate in the hydrogen-bonding interaction with the binding pocket. Furthermore, Chen and coworkers observed that appropriately reducing the length of the alkyl chain at position C3 modestly increased activity.

In contrast, the introduction of alkyl chains at the C6 position significantly enhanced potency. It should be noted that several of the derivatives synthesized by Chen et al. displayed superior activity against PAI-1 compared to embelin. For example, 2-butyl-3,6-dihydroxy-5-octylcyclohexa-2,5-diene-1,4-dione [Figure 1 (**6**)] had an IC_50_ value of 0.18 µM and was 27 times more potent than embelin [38].

Singh and coworkers synthesized embelin derivatives to address the issue of poor aqueous solubility [37]. To achieve this, they introduced nitrogen-containing heterocycles into the embelin scaffold and prepared hydrochloric acid salts of the derivatives. The Mannich reaction was used to introduce the N-linked functionalities to increase the hydrophilicity of embelin. Singh et al. reported a synthetic method for the hydrochloric acid salts of the embelin derivatives shown in Figure 6. Briefly, embelin (**1**) was treated with formaldehyde and a variety of amines in methanol to give compound **23**. The hydrochloric salts of the derivatives were synthesized by passing HCl gas through the reaction mixture to furnish compound **24**.

Singh and coworkers evaluated the in vitro antiproliferative activity of the embelin derivatives in HCT-116 (colon), MCF-7 (breast), MIAPaCa-2 (pancreatic), and PC-3 (prostate) cancer cells. Both compound **7** and its non-salt form were more potent than embelin in HCT-116 cells with IC_50_ values of 29 µM and 30 µM, respectively. The non-salt form of compound **7** also induced apoptotic cell death and caused loss of mitochondrial membrane potential in HCT-116 cells. Out of the fourteen derivatives synthesized, [Figure 1 (**7**)] had an aqueous solubility greater than 1500 µg/mL and was at least 300 times more soluble than embelin. Compound **7** was also more potent than embelin in inhibiting cell proliferation in three of the four cancer cells tested.

The synthetic approaches from the various groups discussed above have led to many embelin derivatives. In addition, promising lead derivative molecules exhibit superior biological activity and improved aqueous solubility compared to embelin. However, more studies are needed to gain deeper insights into the structure–activity relationships. Furthermore, additional research needs to be performed to elucidate the precise mechanism of action of the various embelin derivatives.

## 3. Embelin Delivery Systems

Poor aqueous solubility is one reason limiting the application of embelin for cancer therapy. Consequently, various drug delivery platforms have been explored to improve the aqueous solubility of embelin as well as its stability and bioactivity. This section briefly discusses how polymeric micelles, hydrogels, and polymeric nanoparticles can potentially improve the efficacy of embelin.

### 3.1. Polymeric Micelles

Polymeric micelles are nanosized drug delivery platforms that are formed by the self-assembly of amphiphilic copolymers into a core-shell structure with a hydrophobic core [44]. The hydrophobic core can load hydrophobic drugs, while the hydrophilic polyethylene oxide (PEO) corona provides steric stabilization [44,45]. Additionally, the hydrophilic corona furnishes micelles with stealth properties that allow it to avert micelle aggregation, hinder plasma protein adsorption, and avoid recognition by the reticuloendothelial system (RES), as well as reduce the rapid elimination of micelles from the bloodstream. Furthermore, the small size of micelles guarantees preferential accumulation into tumor cells via the enhanced permeability and retention (EPR) effect [46,47]. Chemical conjugation and encapsulation (physical entrapment) are two approaches by which polymeric micelles can enhance drug solubility [44,48].

Chemical conjugation typically results in high drug loading since the drug is chemically grafted to the copolymer. However, this process requires the presence of super active functional groups such as an amino or a carboxylic acid group on the therapeutic agent, and not all therapeutic agents possess these functional groups. Furthermore, most therapeutic agents cannot be modified without negatively impacting the pharmacological effect. Hence, chemical conjugation is not appropriate for all drugs. In contrast, encapsulation is amenable to a wide range of drugs since it relies on drug compatibility with the hydrophobic core. Although the physics of drug encapsulation in polymeric micelle can be complex, core-forming block length and chemical structural similarity between the hydrophobic core and drug are crucial to high encapsulation [44]. To expedite the clinical translation of polymeric micelles, high encapsulation efficiency, and drug loading is desired regardless of the underlying mechanism used to load drugs into polymeric micelles. Both high encapsulation efficiency and high drug loading translate into using lesser quantities of polymer and lower processing costs.

To date, two groups have spearheaded research on using polymeric micelles to improve the anticancer properties of embelin. In the following, we briefly discuss the contributions from Mahato’s group and Li’s group regarding micellar delivery of embelin (Table 1) [49,50,51,52,53].

#### 3.1.1. Encapsulation

Danquah et al. pioneered the encapsulation of embelin in polymeric micelles for cancer therapy [53]. Embelin-loaded micelles were fabricated by the film sonication method using polyethylene glycol–polylactic acid (PEG-b-PLA) copolymer [Figure 2a], yielding sizes ranging from 30 to 50 nm. To determine the ability of PEG-b-PLA to improve aqueous solubility, they investigated the potential correlation between theoretical drug loading and solubility. As the theoretical drug loading increased from 1 to 20% *w*/*w*, they observed a linear relationship between the amounts of embelin encapsulated into the micelles. Furthermore, they observed a 60-fold increase in aqueous solubility in the micellar system compared to the free drug. The observed improvement in water solubility was hypothesized to be due to better hydrogen bonding between the carbonyl group of the PLA hydrophobic core-forming block and the hydroxyl groups of embelin. The embelin-loaded micelles were used in sequential combination therapy with bicalutamide-loaded micelles to treat LNCaP prostate cancer xenografts. Tumor regression was initially more significant following treatment with bicalutamide-loaded micelles compared to free bicalutamide. After up to 20 days, the tumors became insensitive to bicalutamide and began to grow. However, subsequent treatment with embelin-loaded micelles led to regression of the hormone-insensitive tumors. Therefore, the authors demonstrated the potential utility of bicalutamide- and embelin-based combination therapy for treating advanced prostate tumors and metastases.

Li et al. designed and synthesized a copolymer chemically tailored to enhance the solubility of embelin in polymeric micelles [52]. In their study, they grafted a dodecanol lipid chain onto a poly(ethylene glycol)-b-polycarbonate (PEG-b-PBC) copolymer [Figure 2b] backbone and examined the effect of different chain lengths on loading efficiency. They observed an increase in embelin loading efficiency from 40% in PEG-b-PBC micelles to 100% for poly(ethylene glycol)-block-poly(2-methyl-2-carboxylpropylene carbonate-graft-dodecanol) (PEG-b-PCD) lipopolymer [Figure 2c] at theoretical loading of 5%. This improvement in drug loading was attributed partly to the similarity in structure between embelin and the lipid-modified hydrophobic core of the PEG-b-PCD copolymer [52]. Additionally, the PEG-b-PCD lipopolymer improved the thermodynamic and kinetic stability of the polymeric micelles. This enhancement in micellar stability was hypothesized to be due to the lipid chains in the PEG-b-PCD lipopolymer forming crosslinked structures based on physical entanglement, which improves the hydrophobic interaction in the hydrophobic core. Furthermore, they demonstrated that embelin-loaded PEG-b-PCD micelles significantly inhibited C4-2 prostate cancer cell proliferation.

Polyethylene glycol-b-poly (carbonate-co-lactide) (PEG-b-*p*(CB-co-LA)) copolymer [Figure 2d] has also been used to improve embelin aqueous solubility through encapsulation [51]. Danquah and coworkers observed an encapsulation efficiency of 38.4 ± 1.0% for embelin at 5% theoretical loading and 35.0 ± 0.9% at 10% theoretical loading [51]. Embelin-loaded PEG-b-*p*(CB-co-LA) micelles were used in combination with CBDIV17-loaded PEG-b-*p*(CB-co-LA) micelles to treat C4-2 prostate cancer tumors in xenografts. The combination of CBDIV17 and embelin using a dose of 10 mg/kg more potently inhibited tumor growth compared to the mice treated with empty micelles or monotherapy (Figure 3). Through several studies, Mahato’s group demonstrated the potential of improving the anticancer effect of embelin through encapsulation in various copolymers [51,52,53]. Importantly, their work also elucidated material design rules that can be used to develop next-generation copolymers chemically tailored to significantly improve the aqueous solubility and stability of embelin and ultimately improve its anticancer efficacy.

#### 3.1.2. Conjugation

Using a different approach, Li’s group improved the aqueous solubility of embelin by conjugating it to PEG 3.5K [50]. Although PEG–embelin conjugates can be obtained by directly coupling embelin to PEG using an ester linkage, such an approach results in the PEG randomly attaching to the hydroxyl groups in the benzene ring, which then leads to a mixture of products. Instead, Huang and coworkers developed a total synthesis strategy for conjugating embelin to PEG 3.5K. Their method modifies the total embelin synthesis scheme reported by Wang’s group [42]. It involves conjugating two embelin molecules to one PEG molecule using an aspartic acid linker [42,50]. This synthetic approach results in good yields of well-defined PEG_3.5K_–embelin conjugates [Figure 2e] where the hydroxyl group at the one position of the quinone ring is conjugated to PEG.

Furthermore, the PEG_3.5K–_embelin conjugates had an aqueous solubility of greater than 200 mg/mL, were capable of self-assembling into micelles, and effectively solubilizing hydrophobic anticancer drugs such as paclitaxel. In addition, PEG_3.5K_–embelin micelles exhibited a similar cytotoxic effect to free embelin in MDA-MB-231 human breast cancer cells, 4T1 murine breast cancer cells, and PC3 and DU145 human prostate cancer cells with low μM IC_50_ values. However, PEG_3.5K_–embelin in combination with paclitaxel resulted in synergistic antitumor activity at nM level concentrations in all the four cell lines studied.

In a subsequent study, Li’s group demonstrated that conjugating two molecules of embelin to PEG 3.5K was more effective than conjugates based on a 1:1 molar ratio of PEG and embelin [49]. Furthermore, they showed that the longer chain PEG 5K performed better than the PEG 3.5K. However, in both cases, the PEG–embelin conjugates maintained embelin activity and had a small micelle size (20–30 nm) with minimal effect on micelle size following paclitaxel loading.

Biological evaluations showed PEG 5K–embelin micelles loaded with paclitaxel exhibited an excellent maximum tolerated dose (MTD) of 100–120 mg of paclitaxel per kg. This is greater than the 15–20 mg of paclitaxel per kg reported for Taxol. Compared to Taxol, this increase in MTD increases antitumor activity in breast and prostate cancer mouse models [49].

Since the PEG–embelin conjugates can form micelles, they have the potential to co-deliver a variety of anticancer agents that can facilitate embelin-based combination therapy. Findings reported by Li’s group suggest more structure–activity relationship studies based on their synthesis scheme would be advantageous. Insights obtained may lead to more optimized delivery systems. For instance, stimuli-sensitive functional groups can be included to facilitate intracellular embelin release. Adding crosslinking moieties may also improve the stability of micelles formulated using PEG–embelin conjugates.

### 3.2. Hydrogels

Hydrogels are excellent platforms for local drug delivery. However, its application for cancer therapy has been limited by rapid erosion in vivo. In a recent study, Peng and coworkers created an embelin-loaded thermosensitive injectable hydrogel system less susceptible to fast in vivo erosion and tested its efficacy on mouse hepatic cancers [54]. The injectable embelin thermosensitive hydrogel was obtained using a poly(ε-caprolactone-co-1,4,8-trioxa[4.6]spiro-9-undecanone)–poly(ethylene glycol)–poly(ε-caprolactone-co-1,4,8-trioxa[4.6]spiro-9-undecanone) (PECT) amphiphilic triblock copolymer [Figure 2f], which is an aqueous dispersion at room temperature and transforms into a gel once injected in vivo. The hydrophobic portion of the PECT copolymer solubilized embelin at 10% drug loading, and the resulting embelin/PECT nanoparticles had a particle size of approximately 100 nm. Additionally, the PECT thermosensitive hydrogel improved embelin solubility. It also enhanced its cytotoxicity and ability to induce apoptosis. 

### 3.3. Polymeric Nanoparticles

Polymeric nanoparticles are submicron-sized colloidal particles and are an attractive delivery system for small molecules and nucleic acids [56,57]. Xu and coworkers developed a hyaluronic acid (HA)-coated pH-sensitive poly[(1,6-hexanediol)diacrylate-β-5-hydroxyamylamine] (PBAE)-polyethyleneimine (PEI) [Figure 2g] nanoparticles to co-deliver a therapeutic gene pTRAIL and embelin to treat triple-negative breast cancer cells [55]. Each component of the HA-PBAE-PEI amphiphilic copolymer was selected to provide specific functionality. Specifically, PEI was chosen to condense the pTRAIL gene, and PBAE provided sufficient cargo space to encapsulate embelin. In contrast, HA was selected to make the delivery system site-specific, thereby improving the targeting effect and mitigating the systemic toxicity caused by PEI.

Xu et al. characterized HA-PBAE-PEI nanoparticles and observed monodispersed spherical particles with a core-shell structure having an average particle size of 182.7 ± 2.7 nm. Encapsulation efficiency of embelin was reported to be 57.92 ± 1.14%, while drug loading was 7.24 ± 0.93%. The encapsulation efficiency value for the HA-PBAE-PEI nanoparticles is higher than what was reported by Mahato’s group for the embelin-loaded PEG-b-*p*(CB-co-LA) micelles, which was 38.4 ± 1.0% [51]. This is due to the HA-PBAE-PEI nanoparticles having a more hydrophobic core and hence relatively larger cargo space compared to PEG-b-*p*(CB-co-LA) micelles. It is noteworthy that the HA-PBAE-PEI nanoparticles were more stable than PBAE-PEI nanoparticles in 10% FBS. Furthermore, HA-PBAE-PEI nanoparticles showed a superior cellular uptake in MDA-MB-231 cells, which has higher levels of CD44 compared to MCF-7 cells confirming the enhanced targeting effect of the HA-PBAE-PEI nanoparticles. Moreover, treatment of MDA-MB-231 cells with embelin-loaded HA-PBAE-PEI nanoparticles resulted in a concentration-dependent cytotoxic effect. In addition, it significantly decreased MDA-MB-231 cell viability compared to embelin-loaded PBAE-PEI nanoparticles with no targeting ligand.

## 4. Conclusions and Future Direction

Embelin is a naturally occurring benzoquinone that inhibits the proliferation of several types of cancer cells, making it an effective anticancer agent. However, embelin’s poor aqueous solubility has limited its clinical application. Synthetic strategies such as introducing nitrogen-containing heterocycles into the embelin scaffold or altering the polarity of the linear chain of embelin have been explored to enhance its solubility. Although these approaches have resulted in more water-soluble derivatives, most of them are inactive in cancer cells. Hence, structure–activity relationship strategies using embelin as a pharmacophore for the synthesis of more potent and aqueous soluble derivatives or delivery systems engineered to improve the aqueous solubility of potent but poorly soluble embelin derivatives are needed to facilitate translation from bench-to-bedside. In addition, synthetic strategies that go beyond modification of the hydrophobic tail of embelin need to be explored. Furthermore, increasing evidences in the literature show combining embelin and embelin derivatives with other therapeutic molecules results in a synergistic effect on cancer cell proliferation and apoptosis. Therefore, developing delivery systems that can co-deliver embelin derivatives and other small molecules or embelin derivatives and nucleic acid could be crucial for facilitating cancer therapy.

## Data Availability

Data sharing not applicable.

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
