# Peer review of "Embelin and Its Derivatives: Design, Synthesis, and Potential Delivery Systems for Cancer Therapy"

_pharmaceuticals, 2022, doi:10.3390/ph15091131_

Round 1
Reviewer 1 Report
The manuscript by Danquah on embelin and its derivatives, and concerning an overview of the aspects of design, synthesis and delivery system, is interesting and on the whole well done. However, some important changes are necessary both for the numerous IUPAC nomenclature errors relating to the derivatives described, and for the need to make the article more readable by adding chemical structures and possibly schemes when the derivatives themselves are described. In my opinion, the paper could be reconsidered after the modifications below.
Overall. Please introduce chemical structures and schemes whenever molecules and synthetic procedures are described in the text.
Line (L)11. Delete “previously” (it is pleonastic).
L35. Please write “(2,5-dihydroxy-3-undecyl-1,4-benzoquinone)”.
L35 (and so on for the other molecules of the figure 1). Write “[Figure 1 (1)]”.
L53. “synthetic” instead of “synthesis”.
L69. Please write “(2,5-dihydroxy-3-{2-[4-(2-m-tolylethyl)phenyl]ethyl}-1,4-benzoquinone)”.
L74. “the authors” instead of “they”.
L75. “amino acid” instead of “amino-acid”.
L97,98 and so on. “hydroxyphenyl”, “dimethoxy”, “benzoate” the first letter lowercase.
L102. dihydro-1H-pyrazolo[1,3-b]pyridine (H and b italic).
L103. pyrazolo[1,3-b]pyridine (b italic).
L106. dihydro-1H-pyrazolo[1,3-b]pyridines (H and b italic).
L106. pyrazolo[3,4-b]pyridines (add - and b italic).
L115,120. {6-hydroxy-3-phenyl-4-(4-(trifluoromethyl)phenyl)-7-undecyl-1H-pyrazolo[3,4-b]quinoline-5,8(4H,9H)-dione} (see above).
L156. “it is briefly discussed” instead of “we briefly discuss”.
L162. Please write “polyethylene oxide (PEO)”.
Table 1 last line. “in vitro” italic.
Table 1 caption and L220. Poly(2-methyl-2-carboxylpropylene carbonate-graft-dodecanol)
Table 1 caption. {(S)-N-[4-cyano-3-(trifluoromethyl)phenyl]-3-[(4-cyanophenyl)(methyl)amino]-2-hydroxy-2-methylpropanamide}
Table 1 caption. Poly{g-caprolactone-co-1,4,8-trioxa[4.6]spiro-9-undecanone)–poly(ethylene glycol)–poly(g-caprolactone-co-1,4,8-trioxa[4.6]spiro-9-undecanone}
Table 1 caption. Poly[(1,6-hexanediol)diacrylate-β-5-hydroxyamylamine]
L230. Polyethylene glycol-b-poly(carbonate-co-lactide) [PEG-b-p(CB-co-LA)]
L290. poly(É›-caprolactone-co-1,4,8-trioxa[4.6]spiro-9-undecanone)–poly(ethylene glycol)–poly(É›-caprolactone-co-1,4,8-trioxa[4.6]spiro-9-undecanone)
L314,316. PEG-b-p(CB-co-LA)
L301. poly[(1,6-hexanediol)diacrylate-β-5-hydroxyamylamine]
L330. “derivatives” instead of “derivates”.
L331. “synthetic” instead of “synthesis”.
References. The references are not written as required by the journal. Please check and modify by using an appropriate program.
English. Check correct use of "which".
Author Response
Kindly find the responses to the comments attached. Thank you.

Reviewer 2 Report
The Manuscript of M. Danquah is related to the different methods of improving the pharmacokinetic profile of the natural benzoquinone derivative, embelin and its analogues. The manuscript is well written. However, some major issues should be modified:
1- Figure 1 should be highly improved; for example, you can write beside every compound the method used for optimizing the pharmacokinetic profile of the embelin analogue.
2- Remove the original compound codes from the manuscript and strict to your new numbers (compounds 1-7).
3- The structure PEG3.5K-embelin conjugate should be added to Figure 1 or a separate Figure.
4- The results of compounds 1-7 needs better representation.
5- In page 3 line 94, the sentence “This approach allowed considerable flexibility in the nature and length of the substituent chain”. Explain how compound 4 has a considerable flexibility.
6- Page 3 lines 112-117. The activity of compound 5 should be compared to embelin.
7- HCl salt of compound 7 should be added to the structure in figure 1.
8- Page 4 lines 172-173, “However, this process requires the presence of super active functional groups on the therapeutic agent, and not all therapeutic agents possess these functional groups”. Give examples of these groups in the text.
9- The best 2-3 strategies represented in the manuscript to enhance the solubility should be added in conclusion.
Finally, this work could be accepted after a major revision.
Author Response

(The authors gave the same response as above.)

Round 2
Author Response
Thank you for the feedback. Please see the point-by-point response attached.

Reviewer 2 Report
I recommend this manuscript for publication.
Author Response
Thank you for your feedback and for recommending the manuscript for publication.